# The Impact of Obesity on Nighttime Blood Pressure Dipping

**DOI:** 10.3390/medicina56120700

**Published:** 2020-12-15

**Authors:** Beata Moczulska, Maciej Zechowicz, Sylwia Leśniewska, Karolina Osowiecka, Leszek Gromadziński

**Affiliations:** 1II Clinic of Cardiology and Internal Medicine, Collegium Medicum, School of Medicine, University of Warmia and Mazury in Olsztyn, 11-041 Olsztyn, Poland; m.zechowicz@gmail.com (M.Z.); sylwialesniewska0@gmail.com (S.L.); lgol@op.pl (L.G.); 2Department of Public Health, Unit of Public Health, University of Warmia and Mazury in Olsztyn, 11-041 Olsztyn, Poland; k.osowiecka86@gmail.com

**Keywords:** obesity, ABPM, hypertension

## Abstract

*Background and objectives:* It is commonly known that obesity not only increases arterial hypertension (HT) risk but also impacts on the response to antihypertensives. This study aimed to assess blood pressure (BP) parameters based on Ambulatory Blood Pressure Measurement (ABPM) in obese patients. *Materials and Methods:* The study group consisted of 128 patients with obesity (BMI ≥ 30 kg/m^2^), with an average age of 43.25 years (±12.42), including 55 males and 73 females. They were divided into 2 groups: 1-with BMI ≥ 30 kg/m^2^ and <40 kg/m^2^, 2-with BMI ≥ 40 kg/m^2^. Each patient underwent 24-h blood pressure monitoring. The average 24-h, daytime and nighttime systolic and diastolic pressure, as well as 24-h mean heart rate and % of nocturnal dip, were assessed. *Results:* Mean BMI in group 1 was 34.73 kg/m^2^ (±2.96), and in group 2 it was 47.6 kg/m^2^ (±6.3). Group 1 was significantly older than group 2 (46.5 vs. 39 years old). The analysis of ABPM revealed significantly higher BP values in all measurements in group 2 (i.e., systolic blood pressure (SBP) 24 h median = 132 mmHg; diastolic blood pressure (DBP) 24 h median = 84 mmHg). The nocturnal dip was greater in group 1 (8.95%). Mean 24-h heart rate was also higher in group 2 (median = 76 beats/min) than group 1 (median = 67.5 beats/min). More than half of patients in group 2 had been previously treated for HT, and based on ABPM, new HT was diagnosed in 6 patients from group 1 and 14 patients from group 2. Three groups of patients were identified based on nighttime dip: dipper, non-dipper, and reverse-dipper. No patient of the extreme dipper type was found. Group 2 comprised of significantly more patients of the reverse-dipper type. *Conclusions*: Patients with extreme morbid obesity frequently exhibit HT of the reverse-dipping pattern. This type is often linked with a higher risk of more advanced cardiovascular illness.

## 1. Introduction

Obesity is one of the most important cardiovascular risk factors. According to the NCD Risk Factor Collaboration the number of worldwide obesity cases has tripled in the last 45 years with over 1.9 billion adults (with 39% overweight and 13% obese) [1]. Over 50% of women are overweight in Europe, and about half of them are obese [2]. In Poland the NATPOL 2011 study confirmed the increase of obese individuals to 22% [3]. It is estimated that by 2030 nearly two-thirds of the global population will be either overweight or obese [4]. The positive correlation between body mass and the development of arterial hypertension (HT) has been evidenced by numerous epidemiological studies [5,6,7]. Presently, it is commonly known that obesity not only increases HT risk but also impacts on the response to antihypertensive drugs. Ambulatory Blood Pressure Measurement (ABPM) seems to be a very useful tool: It allows for a very precise analysis of mean blood pressure (BP) values in 24-h monitoring, but also for assessing the profile of nocturnal BP values as compared to daytime values.

Dipping patterns are classified by the percent of drop in pressure, and based on the resulting ratios, a person may be clinically classified. The extent of the nighttime BP dipping may be a predictor for the prevalence of cardiovascular complications. Reverse-dippers are characterized by the worst cardiovascular prognosis, both as regards stroke and cardiovascular events. ABPM allows for identifying the patient dipping pattern (dippers) and then customizing antihypertensive treatment.

## 2. Patients and Methods

The study group consisted of 128 patients with obesity (BMI ≥ 30 kg/m^2^), with an average age of 43.25 years (±12.42), including 55 males and 73 females, hospitalized in II Clinic of Cardiology and Internal Medicine of the University Clinical Hospital in Olsztyn in the period 2017–2019 to be assessed clinically prior to bariatric surgery. Patients with new infection, fever, cancer, liver diseases and pulmonary diseases were excluded from the study.

The study protocol was approved by the Bioethics Committee at the School of Medicine of the University of Warmia and Mazury in Olsztyn on 22 June 2017. Each patient agreed to participate in the study by signing informed consent. Our research has received approval from an ethics committee: decision No. 28/2017 of the Bioethics Committee at the University of Warmia and Mazury in Olsztyn. A detailed medical interview was conducted with each patient including antihypertensive medications. The drugs were divided into 5 groups: β-blocker, ACE-I/ARB, diuretic, calcium channel blocker and other (centrally acting antihypertensive, α-blocker).

Body Mass Index (BMI) was derived from Quetelet’s equation: body mass (kg)/height (m^2^). Obesity was diagnosed based on the BMI value consistent with the World Health Organization (WHO) criteria, according to which obesity is classified as a BMI ≥ 30.0 kg/m^2^ [8]. Patients were divided into 2 groups:Group 1: patients with BMI ≥ 30 kg/m^2^ and <40 kg/m^2^;Group 2: patients with BMI ≥ 40 kg/m^2^.

Laboratory tests were performed for all patients, including the assessment of C-reactive protein (CRP), glucose, liver enzymes, thyroid-stimulating hormone (TSH), creatinine, uric acid and lipid profile.

For each patient, 24-h BP monitoring was performed (ABPM) with the Mobil-O-Graph NG PWA device by the IEM company (Germany). Before fitting the device, BP was measured on both upper limbs for each patient. The measurement was taken in the sitting position after a minimum of 5-min rest. When both differences between systolic blood pressure (SBP) and diastolic blood pressure (DBP) measured on the upper extremities were less than 10 mmHg, the cuff was put on the dominating extremity. When BP difference was higher than 10 mmHg, the cuff was put on the arm with a higher BP value. The cuff size was adjusted according to the arm circumference according to the ABPM protocol. The device recorded SBP, DBP and heart rate (HR) automatically every 30 min at nighttime (22.00–6.00) with a silent mode, and every 15 min during the day (6.00–22.00), indicating the upcoming measurement with a sound signal. HT was diagnosed based on 24-h BP values > 135/85 mmHg consistent with the European Society of Hypertension (ESH) guidelines [9]. The device was fitted during the 1st or 2nd day of hospitalization.

The following parameters were assessed: mean 24-h, daytime and nighttime systolic and diastolic pressure, as well as mean 24-h heart rate and the extent of nocturnal BP dip. Patients with normal nighttime dip (by 10–20% as compared to daytime values) were termed dippers, with nighttime dip <10%-non-dippers, with night-time dip >20%-extreme dippers, and when BP values during the sleep were higher than those during the non-sleep period—reverse-dippers.

Depending on the obtained medical interview and the ABPM result, we identified patients with treated HT, newly detected HT and without HT.

## 3. Statistical Analysis

Statistical analysis was performed with Statistica 13.1 (Statsoft, Poland) software. The descriptive statistics of parameters (median, 25–75% quartile, mean, standard deviation, percentages) were used in the analysis. The distributions of continuous variables were compared with the theoretical normal distribution using the Shapiro-Wilk test. The distribution of variables differed significantly from normal (*p* < 0.05), so the nonparametric tests were used. The differences in age, laboratory parameters and ABPM parameters between two subgroups (BMI < 40 vs. ≥40) were analyzed with the Mann-Whitney test. A comparison of the proportion (arterial hypertension, taken antihypertensive drugs, nocturnal blood pressure dip) in subgroups was tested using the chi-square test. A *p*-value of <0.05 was considered to be significant.

## 4. Results

Based on BMI, patients were divided into 2 groups: Group 1, patients with BMI ≥ 30 kg/m^2^ and <40 kg/m^2^, and Group 2, patients with BMI ≥ 40 kg/m^2^. Mean BMI in group 1 amounted to 34.73 kg/m^2^ (±2.96), and in group 2, it was 47.6 kg/m^2^ (±6.3). An average age of study patients was 43.25 years (±12.42). Group 1 was significantly older than group 2 (46.5 vs. 39 years old).

The analysis of the lipid profile revealed abnormalities for high-density lipoprotein (HDL) levels, which were significantly higher in group 1, and for triglycerides (TG), which were higher in group 2. CRP levels were higher in group 2 (2.45 vs. 5.95), and creatinine levels were higher in group 1. Uric acid levels were higher in group 2 but without statistical significance, similarly to TSH and liver enzymes (Table 1).

The analysis of ABPM revealed that all BP values were significantly higher in group 2. The nocturnal dip was greater in group 1, but without statistical significance. The mean 24-h heart rate was also higher in group 2 (see Table 2 for details).

Based on the analysis of nocturnal BP dip, 3 groups of patients were identified: dipper, non-dipper and reverse-dipper. No patient of the extreme dipper type was found. Group 2 comprised of significantly more patients of the reverse-dipper type (see Figure 1).

More than half of the patients in group 2 had been previously treated for HT (53.2%), 47.2% of patients from group 1 had antihypertensive treatment. The most frequently administered drugs in both groups were ACE-I or ARB. The second most-often taken drug was β-blocker in group 1 and diuretic in group 2 (see Table 3 for details).

Based on ABPM, new HT was detected in 6 patients from group 1 (17.6%) and 14 (14.9%) patients from group 2 (*p* = 0.7). One-third of all patients were not treated for HT (see Figure 2).

## 5. Discussion

Monitoring BP with ABPM is central not only for diagnosing HT but also for its adequate treatment. The benefit of ambulatory 24-h monitoring over clinical measurements was also evidenced in the context of mortality risk assessment [10]. ABPM plays a special role in the treatment of obese patients, in whom a significant clinical association between nocturnal change in BP and the risk of cardiovascular events was demonstrated [11,12]. Of significance are also correlations between BP values, BMI and the detrimental changes in cardiac structure, such as myocardial hypertrophy and left atrial enlargement, which may indicate early stages of elevated left ventricular filling pressure [13,14]. In the study group of 128 patients with obesity and BMI of at least 30 kg/m^2^, who had been considered as candidates for bariatric surgery, 2 sub-groups were identified: patients with class 1 and 2 obesity (group 1 with BMI 30–39 kg/m^2^) and patients with class 3 obesity (group 2 with BMI ≥ 40 kg/m^2^). First of all, it is worth noting that the average age in group 1 was statistically significantly higher than in group 2 (46.5 and 39 years, *p* = 0.004, respectively); moreover, in group 2 no patient older than 50 years old was found. Additionally, these groups differed in terms of metabolism: group 1 had statistically significantly higher HDL and lower TC levels (51.5 mg/dL and121 mg/dL vs. 44 mg/dL and 147 mg/dL, *p* = 0.05, respectively). This tendency was consistent with the observations of other researchers dealing with the qualification of patients for bariatric procedures [14,15]. According to an interesting meta-analysis, metabolic syndrome, including all or almost all its potential components (4 or 5 out of 5), is associated with earlier and more serious organ damage, at both cardiac and vascular levels [16]. Although both analyzed groups were quite similar in terms of the occurrence of HT and its current treatment, including the administered antihypertensives, the ABPM analysis demonstrated that in the group of patients with BMI ≥ 40 kg/m^2^ all BP values were higher than in the less obese group.

According to D’Elia et al. overweight or obesity and accompanying insulin resistance could mediate the development of isolated systolic hypertension through the increase in the renin-angiotensin-aldosterone system activity, in the sympathetic tone and salt-sensitivity, all in turn leading to endothelial dysfunction, arterial stiffness and increase in blood pressure [17].

Interestingly, it was observed that in the more obese group, the diastolic pressure increases more (*p* < 0.001) than systolic pressure as compared to the less obese group (e.g., mean 24-h systolic pressure was higher in group 2 by 5 mmHg as compared to group 1, and diastolic pressure by as much as 9 mmHg). A similar association, additionally correlated with increased vessel stiffness in such patients, was reported by Korean authors, though their patient group was rather small [18]. Unlike in our study, according to the paper by Higgins et al. of 1988, based on the Framingham study, greater body weight was associated with increased systolic and diastolic pressure proportionally to baseline—patients with the upper quintile values of BMI demonstrated higher systolic pressure by 16 mmHg and higher diastolic pressure by 9 mmHg as compared to patients with the lowest quintile values of BMI [19]. It must be stressed, however, that the observed statistically significantly higher diastolic pressure in the most obese patients requires further clinical and scientific analyses, especially since there is little data on the long-term prognosis of such patients. It should be assumed that increased diastolic pressure in patients with pathological obesity is significantly—clinically associated with the pathological alteration in cardiac structure in such patients, which may eventually lead to the development of heart failure with preserved ejection fraction and worse prognosis [20]. This correlation and a more negative cardiovascular prognosis for patients with pathological obesity may be supported by the analysis of nocturnal BP dipping in the two studied groups. Although statistically decreased BP by 8.95 mmHg in group 1 (Median, 5.3–11.4 (25–75% quartile)) and by 7.4 mmHg in group 2 (Median, 1.2–12.4 (25–75% quartile)), *p* = 0.42, during nighttime hours reported by patients as sleep-time, in group 1, only 2 patients qualified as reverse-dippers, and in group 2, as many as 20 such patients were identified (5.9% and 21.3%, *p* = 0.04, respectively). The number of dippers was similar (35.3% in group 1 and 35.1% in group 2, *p* = 0.98, respectively). Slightly more often, non-dippers were found in group 1 as compared to group 2 (58.8% vs. 43.6%, respectively). However, this correlation was not statistically significant (*p* = 0.13) and was associated with a significantly greater number of patients of the reverse-dipper type in group 2. Obesity itself is claimed to be one of the central causes of the non-dipping phenomenon (together with endocrine disorders, kidney dysfunction, and autonomic dysfunctions); however, there is scarce data concerning reverse-dippers. Based on the Korean population, Kim et al. demonstrated, that reverse-dipping is a significant cause of cardiovascular mortality [21]. In another interesting study from Asia, authors again refer to abnormal cardiac structural alterations in hypertensive patients of the non-dipper type as a risk factor for infarctions and strokes [22]. Similar conclusions may be drawn based on a large meta-analysis comprising of over 17 thousand patients from 3 different continents, in which a significantly greater risk of total cardiovascular events, but also cardiovascular mortality and total mortality were observed in reverse-dippers as compared to dippers. Interestingly, the risk was comparable to that of untreated hypertensive patients (HR 1.57–1.89 vs. 1.92) [23]. Earlier, Fagard et al., in their meta-analysis of the 24-h BP pattern in more than three thousand hypertensive patients, also demonstrated that reverse-dippers are at the greatest risk of cardiovascular events; however, analyses of the sub-groups concerning BMI were not available [24]. At the same time, it was reported, that in repeated ABPM studies the reverse-dipper type is not always maintained in subsequent assessments and the factor most strongly associated with the increased risk of cardiovascular events is increased systolic pressure at night regardless of “dipping”, as found in patients with diabetes mellitus and hypertension [25]. An abnormally high level of catecholamines in such patients and the correlation of catecholamines with an increased risk of cardiovascular complications are suggested as one of the reasons for this phenomenon [26]. In our population, the factor indicating an increased adrenergic stimulation in the more obese group is the increased heart rate: patients in group 2 had higher HR (Median = 76 beats/min; 69.0–81.0 (25–75% quartile)) than patients in group 1 (Median = 67.5 beats/min, 63.0–75 (25–75% quartile)), this correlation being statistically significant at *p* < 0.001. The impact of obstructive sleep apnea on the obtained results was excluded (detected in 11.8% of patients in group 1 and 7.4% in group 2, *p* = 0.68). Additionally, a possible impact of sleeping pills on nighttime measurements was also excluded (only 2 patients in group 2 took a sleep-inducing drug).

This study had a number of limitations such as small sample size. Future investigation will aim to include a larger and more diverse sample. Moreover, a long-term observation outside of hospital would allow better understanding of the HT impacts and treatments as well as linkage to other conditions. Furthermore, the ABPM assessment was based only on one measurement. Ideally, this would be done multiple times and using different equipment to confirm validity. Furthermore, the ABPM measure should be taken outside of the hospital setting to ensure it reflects the patient’s lifestyle, but at the same time, the measurement conditions were very similar for all of the patients included in the study.

The presented analyses and available literature do not offer exact assessments of cardiovascular events and mortality risk in this particular population of patients with pathological obesity and HT characterized by reverse-dipping. However, based on numerous publications that refer to these risk factors separately, but also in the context of the consistency of the results of the conducted analyses, it may be assumed with a great likelihood that the most obese patients who are reverse dippers will constitute the greatest risk group. Therefore, ABPM plays a central role in the diagnosing and careful monitoring of HT as well as for the prevention of cardiovascular and cerebral events, which are as likely as if we completely ignored and not treated HT in these patients.

## 6. Conclusions

Patients with morbid obesity have most often HT characterized by the reverse-dipping pattern. HT of such a profile is associated with greater cardiovascular risk. It is justifiable then to perform ABPM for all obese patients in order to determine the type of HT and to customize antihypertensive therapy, especially, to intensify treatment in the evening.

## Figures and Tables

**Figure 1 medicina-56-00700-f001:**
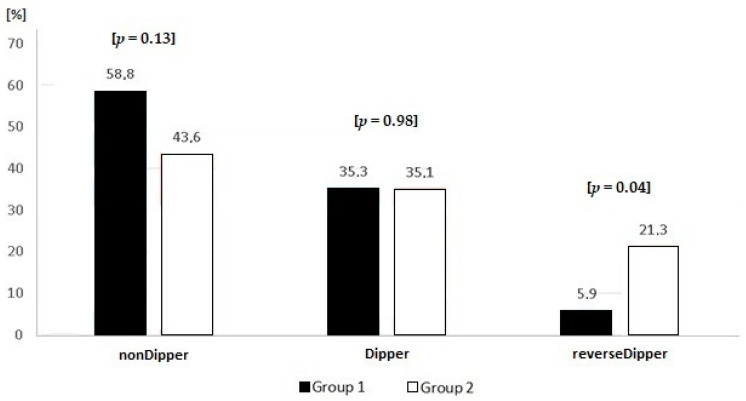
Patients according to nocturnal blood pressure dipping.

**Figure 2 medicina-56-00700-f002:**
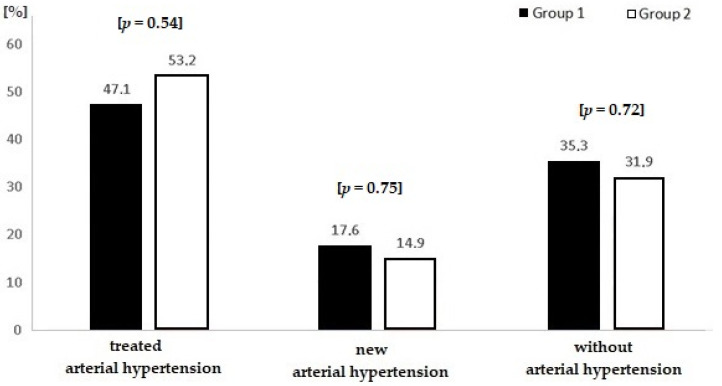
Arterial hypertension in both groups.

**Table 1 medicina-56-00700-t001:** Laboratory parameters.

Parameter	Groups				
	Group 1 < 40 (*n* = 34)		Group 2 ≥ 40 (*n* = 94)		
*n* = 128	median	(25–75% quartile)	median	(25–75% quartile)	*p*
Age (years old)	46.5	(39.0–58.0)	39.0	(33.0–48.0)	0.004
Total cholesterol (mg/dL)	196.5	(177.0–212.0)	192.0	(167.0–211.0)	0.43
Low-density lipoprotein (mg/dL)	119.5	(97.0–133.0)	115.5	(86.0–131.0)	0.52
High-density lipoprotein (mg/dL)	51.5	(43.0–62.0)	44.0	(40.0–51.0)	0.006
Triglycerides (mg/dL)	121.0	(105.0–162.0)	147.0	(119.0–200.0)	0.05
Uric acid (mg/dL)	5.85	(5.4–6.9)	6.25	(5.6–7.1)	0.23
C-reactive protein (mg/L)	2.45	(1.0–5.3)	5.95	(3.6–8.8)	0.0003
Aspartate transaminase (U/L)	22.5	(18.0–28.0)	22	(19.0–28.0)	0.83
Alanine transaminase (U/L)	27.5	(19.0–37.0)	28.0	(23.0–42.0)	0.55
Thyroid-stimulating hormone (µIU/mL)	1.65	(1.19–2.53)	1.59	(1.19–2.15)	0.58
Glucose (mg/dL)	95.0	(89.0–102.0)	92.5	(88.0–113.0)	0.85
Creatinine (mg/dL)	0.77	(0.68–0.87)	0.65	(0.61–0.79)	0.002

**Table 2 medicina-56-00700-t002:** Ambulatory Blood Pressure Measurement (ABPM) parameters.

Parameter	Groups				
	Group 1 < 40 (*n* = 34)		Group 2 ≥ 40 (*n* = 94)		
*n* = 128	median	(25–75% quartile)	median	(25–75% quartile)	*p*
Systolic blood pressure 24-h (mmHg)	127.0	(116.0–134.0)	132.0	(127.0–140.0)	0.004
Diastolic blood pressure 24-h (mmHg)	75.0	(72.0–82.0)	84.0	(76.0–90.0)	0.0003
Systolic blood pressure daytime (mmHg)	129.5	(119.0–138.0)	134.0	(128.0–141.0)	0.008
Diastolic blood pressure daytime (mmHg)	77.5	(74.0–86.0)	86.0	(78.0–93.0)	0.0007
Systolic blood pressure nighttime (mmHg)	117.0	(108.0–126.0)	124.0	(117.0–136.0)	0.003
Diastolic blood pressure nighttime (mmHg)	68.5	(63.0–74.0)	75.5	(66.0–83.0)	0.002
Dipping %	8.95	(5.3–11.4)	7.4	(1.2–12.4)	0.42
Heart rate 24-h (beats per minute)	67.5	(63.0–72.0)	76.0	(69.0–81.0)	0.00005

**Table 3 medicina-56-00700-t003:** Taken antihypertensive drugs.

Drug	Groups				
	Group 1 < 40 (*n* = 34)		Group 2 ≥ 40 (*n* = 94)		
*n* = 128	*n*	%	*n*	%	*p*
β-blocker	14	41.2	27	28.7	0.18
ACE-I/ARB	21	61.8	49	52.1	0.33
Diuretic	13	38.2	33	35.1	0.74
Ca-blocker	12	35.3	31	33	0.81
Other	1	2.9	4	4.3	0.86

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
