# Peer review of "The Impact of Obesity on Nighttime Blood Pressure Dipping"

_medicina, 2020, doi:10.3390/medicina56120700_

Round 1
Reviewer 1 Report
The objective of this paper is to assess blood pressure (BP) parameters based on Ambulatory Blood Pressure Measurement (ABPM) in obese patients. The study group consisted of 128 patients with obesity (BMI≥30kg/m2), hospitalized to be assessed clinically prior to bariatric surgery. For each patient 24-hour BP monitoring was performed (ABPM) with the Mobil-O-Graph NG PWA device. Laboratory tests were performed for all patients, including the assessment of CRP, glucose, liver enzymes, TSH, creatinine, uric acid and lipid profile. The study group was divided into 2 groups: 1 –with BMI ≥30kg/m2 and <40kg/m2 with BMI ≥40kg/m2 . The analysis of ABPM revealed significantly higher BP values in all measurements in group 2. The nocturnal dip was greater in group 1. Mean 24-hour pulse was also higher in group 2. More than half
of patients in group 2 had been previously treated for HT and based on ABPM new HT was
28 diagnosed in 6 patients from group 1 and 14 patients from group 2. Three groups of patients were identified based on nighttime dip: dipper, non-dipper, and reverse-dipper. No patient of the extreme dipper type was found. Group 2 comprised of significantly more patients of the reverse-dipper type.
This study reaffirms some observations already demonstrated in previous large studies (1)
Mayor comments:
-One of the main problems is that abmp measurements have been done in hospitalized subjects. As stated in the European Society of Hypertension Position Paper on Ambulatory Blood Pressure Monitoring “ABPM is best not performed while patients are admitted to hospital because hospitalization leads to underestimation of BP and over- diagnosis of nondippers”. So this condition could bias the results.
-Clinical blood pressure values are not included in the study. The comparison of clinical with abpm values would be interesting. Taking into account this information, white coat and masked hypertension phenomena could be analyzed.
-I wonder if results have been controlled by variables as biochemical measurements or if some relation among this data a different BP profiles has been found.
-It has been described and increase
-ABMP device used by researchers is able to calculate pulse wave velocity. Has been considered by authors to include that information in this paper to increase novelty?.
-The paper should include the quality of ABMP measurements. Taking into account the difficulty of making this type of measurements in extreme obese subjects, Which were the criteria to consider the recordings as acceptable?
Minor comments
-In the introduction, data of obesity prevalence and references could be updated.
-In page 1 and 7, at the end there is a reference that may be suppressed.
-In statistical analysis, methods used for the analysis continuous variables should be described. This paragraph should be reviewed.
-Along the text it is used the term “reverse dipper”, while in figure 1 it is used “Riser”
-In line 119 it is said “Based on ABPM, new HT was detected in 6 patients from group 1 (17.6%) and 14 (14.9%) patients from group 2 (p=0.7).” IN methods should be described how was this diagnosis made. What criteria was used? Where normotensive by clinical BP assessment? How can it be supported if clinical BP is not included in the study? Were these patients masked hypertensive subjects?
-Number of figures are interchanged.
-In figure 1, please include if differences observed are statistically significant.
-In line 195, it is said that “The impact of obstructive sleep apnea on the obtained results
196 was excluded (detected in 11.8% of patients in group 1 and 7.4% in group 2, p=0.68)”. How was SAOS assessed?. Please, his information should be included in results section to be commented in the discussion.
References:
- Williams B, Mancia G, Spiering W, Rosei EA, Azizi M, Burnier M, et al. 2018 practice guidelines for the management of arterial hypertension of the European society of cardiology and the European society of hypertension ESC/ESH task force for the management of arterial hypertension. Vol. 36, Journal of Hypertension. 2018. 2284–2309 p.
Author Response
Dear anonymous reviewer,
Thank you for giving us the opportunity to submit a revised draft of our manuscript titled "The impact of obesity on nighttime blood pressure dipping". We appreciate the time and effort that you have dedicated to reviewing our manuscript. We are grateful for your insightful comments on our paper. We have incorporated changes to address most of your suggestions. Below is a point-by-point response to the comments.
Comment:
-One of the main problems is that abmp measurements have been done in hospitalized subjects. As stated in the European Society of Hypertension Position Paper on Ambulatory Blood Pressure Monitoring “ABPM is best not performed while patients are admitted to hospital because hospitalization leads to underestimation of BP and over- diagnosis of nondippers”. So this condition could bias the results.
Response:
Thank you for pointing this out. We agree with this comment. But our patients traveled from all parts of Poland and we were unable to evaluate the measurement outside the hospital. We took into account that the measurement used in a hospital setting may affect the results
Comment:
-Clinical blood pressure values are not included in the study. The comparison of clinical with abpm values would be interesting. Taking into account this information, white coat and masked hypertension phenomena could be analyzed.
Response:
We agree. It would have been interesting to explore this aspect. We did not consider clinical measurements on admission as the values ​​would be overestimated by the white coat effect, but we will consider analyzing the comparison of clinical BP with ABPM values in the next paper.
Comment:
-I wonder if results have been controlled by variables as biochemical measurements or if some relation among this data a different BP profiles has been found.
Response:
I assume this comment refers to Table 1. In this study, we did not look into more details of controlled variables but our intention is to investigate it in future with the bigger sample size.
Comment:
-It has been described and increase
Response:
Apologies but we do not understand this comment. Please could you clarify and highlight the area it refers to. Thank you.
Comment:
-ABMP device used by researchers is able to calculate pulse wave velocity. Has been considered by authors to include that information in this paper to increase novelty?.
Response:
We are deeply impressed by this observation. Thank you very much. It would have been interesting to include this aspect We will definitely consider your comment and assess pulse wave velocity in our further work.
Comment:
-The paper should include the quality of ABMP measurements. Taking into account the difficulty of making this type of measurements in extreme obese subjects, Which were the criteria to consider the recordings as acceptable?
Response:
We are aware of the challenges of taking this type of measurements. We followed Polish guidance which are based on 2013 European Society of Hypertension [ESH] criteria. They state having ≥70% of planned readings with a minimum of 20 daytime and 7 nighttime readings.
Comment:
-In the introduction, data of obesity prevalence and references could be updated.
Response:
Thank you for pointing this out. We agree with this comment. Therefore, appropriate corrections have been made on page 2 , in the introduction
Comment:
-In page 1 and 7, at the end there is a reference that may be suppressed.
Response:
Changed as suggested.
Comment:
-In statistical analysis, methods used for the analysis continuous variables should be described. This paragraph should be reviewed.
Response:
Changed as suggested.
Comment:
-Along the text it is used the term “reverse dipper”, while in figure 1 it is used “Riser”
Response:
Changed as suggested.
Comment:
-In line 119 it is said “Based on ABPM, new HT was detected in 6 patients from group 1 (17.6%) and 14 (14.9%) patients from group 2 (p=0.7).” IN methods should be described how was this diagnosis made. What criteria was used? Where normotensive by clinical BP assessment? How can it be supported if clinical BP is not included in the study? Were these patients masked hypertensive subjects?
Response:
We answered this point in patients and methods section. We assumed that those readers who will be interested in the details will read ESH guidelines.
Comment:
-Number of figures are interchanged.
Response:
Changed as suggested. We changed both Figures to be better presented.
Comment:
-In figure 1, please include if differences observed are statistically significant.
Response:
Changed as suggested.
Comment:
-In line 195, it is said that “The impact of obstructive sleep apnea on the obtained results
196 was excluded (detected in 11.8% of patients in group 1 and 7.4% in group 2, p=0.68)”. How was SAOS assessed?. Please, his information should be included in results section to be commented in the discussion.
Response:
We did not identify any new obstructive sleep apnea syndromes (OSAS), if someone had already been diagnosed and treated, we included it in our work
We look forward to hearing from you in due time regarding our submission and to respond to any further questions and comments you may have.
Sincerely,
Beata Moczulska, Maciej Żechowicz, Sylwia Leśniewska, Karolina Osowiecka, Leszek Gromadziński

Reviewer 2 Report
This study is interesting and has the potential to contribute to the knowledge of this topic .However, there are numerous factors that confound the overall results such as diet, physical activity, sleep history etc. during the time in which the ABPM was worn. Then there is the issue of having only one measurement and reliable the measure is unknown. Rather than divide into two groups it would be more appropriate to run mixed linear models and possible other types of analyses in which you can determine relationships. There are numerous factors that should be included in analyses as covariates such as medication, sleep apnoea and other medical conditions. Below are my further comments.
Line 14: Spell out HT before abbreviating.
Line 21: 24-hour mean pulse – change to heart rate and not ‘pulse’
Lines 23-24: Units needed for BMI results.
Lines 23-24: Characteristics of participants should be included in Methods section.
Lines 25-26: Provide data on the results of BP and heart rate for the groups and differences.
Line 32: This following sentence is confusing “Arterial hypertension with this development profile develops with a greater cardiovascular event.” – Please revise.
Lines 50-52: Definition of the types of ‘dippers’ is needed.
Line 62: Add reference for WHO criteria.
Lines 66-67: Spell out CRP and TSH before abbreviating.
Line 69: Include specifications of equipment, region country etc.
Line 77: What happened when only SBP or DBP was greater than 135/85mmHg? Isolated hypertension for that particular value?
Lines 80-81: Be consistent with abbreviations e.g. SBP
Lines 92-93: Ethics information should be included in the first paragraph of methods plus information about participant informed consent.
Lines 95-99: Participant characteristics should be included in Methods. Also, make sure you include relevant units when reporting data e.g. BMI.
Table 1 and Table 2: Spell out all abbreviations.
Figure 2: Is poorly presented. Remove lines and colour black and white. Also, removed data above bars. Where are the axis titles and units?
Line 113: Figure 2 label should be Figure 1.
Line 122: Label should be Figure 2. Also see my comments about the other Figure.
Line 167: Me? Please spell out. Also make sure you include units when reporting data.
Line 194: Me? Please spell out. Also make sure you include units when reporting data.
You need to include limitations such as only one ABPM assessment, no documentation on physical activities, diet etc. that may have affect BP throughout the monitoring. Relatively small sample size.
Author Response
Dear anonymous reviewer,
Thank you for giving us the opportunity to submit a revised draft of our manuscript titled "The impact of obesity on nighttime blood pressure dipping". We appreciate the time and effort that you have dedicated to reviewing our manuscript. We are grateful for your insightful comments on our paper. We have incorporated changes to address most of your suggestions. Below is a point-by-point response to the comments. And attached the file with the changed manuscript.
Comment:
Line 14: Spell out HT before abbreviating.
Response:
Changed as suggested.
Comment:
Line 21: 24-hour mean pulse – change to heart rate and not ‘pulse’
Response:
Changed as suggested.
Comment:
Lines 23-24: Units needed for BMI results.
Response:
Changed as suggested.
Comment:
Lines 23-24: Characteristics of participants should be included in Methods section
Response:
Changed as suggested.
Comment:
Lines 25-26: Provide data on the results of BP and heart rate for the groups and differences.
Response:
Changed as suggested.
Comment:
Line 32: This following sentence is confusing “Arterial hypertension with this development profile develops with a greater cardiovascular event.” – Please revise.
Response:
We changed the sentence to: “This type of HT is often linked with a higher risk of greater cardiovascular illness. “
Comment:
Lines 50-52: Definition of the types of ‘dippers’ is needed.
Response:
Updated.
Dipping patterns are classified by the percent of drop in pressure, and based on the resulting ratios a person may be clinically classified. The extent of nighttime BP dipping may be a predictor for the prevalence of cardiovascular complications. Reverse-dippers are characterized by the worst cardiovascular prognosis, both as regards stroke and cardiovascular events. ABPM allows for identifying the patient dipping pattern (dippers) and then customizing antihypertensive treatment.
Comment:
Line 62: Add reference for WHO criteria.
Response:
Changed as suggested.
Comment:
Lines 66-67: Spell out CRP and TSH before abbreviating.
Response:
Changed as suggested.
Comment:
Line 69: Include specifications of equipment, region country etc.
Response:
No more details provided
Comment:
Line 77: What happened when only SBP or DBP was greater than 135/85mmHg? Isolated hypertension for that particular value?
Response:
We did not assess and recognize isolated hypertension, there was no such case in our group of subjects
Comment:
Lines 80-81: Be consistent with abbreviations e.g. SBP
Response:
Changed as suggested.
Comment:
Lines 92-93: Ethics information should be included in the first paragraph of methods plus information about participant informed consent.
Response:
Changed as suggested.
Comment:
Lines 95-99: Participant characteristics should be included in Methods. Also, make sure you include relevant units when reporting data e.g. BMI.
Response:
Changed as suggested.
Comment:
Table 1 and Table 2: Spell out all abbreviations.
Response:
Changed as suggested.
Comment:
Figure 2: Is poorly presented. Remove lines and colour black and white. Also, removed data above bars. Where are the axis titles and units?
Response:
We are deeply impressed about this observation and changes. Thank you very much. Changed as suggested.
Comment:
Line 113: Figure 2 label should be Figure 1.
Response:
Changed as suggested.
Comment:
Line 122: Label should be Figure 2. Also see my comments about the other Figure.
Response:
Changed as suggested.
Comment:
Line 167: Me? Please spell out. Also make sure you include units when reporting data.
Response:
Changed as suggested.
Comment:
Line 194: Me? Please spell out. Also make sure you include units when reporting data.
Response:
Changed as suggested.
Comment:
You need to include limitations such as only one ABPM assessment, no documentation on physical activities, diet etc. that may have affect BP throughout the monitoring. Relatively small sample size.
Response:
Thank you very much for the thorough evaluation of the article. It would have been interesting to explore this aspect. We were limited by our available sample. We have added a subsection dedicated to the study limitations at the end of the discussion section.
We look forward to hearing from you in due time regarding our submission and to respond to any further questions and comments you may have.
Sincerely,
Beata Moczulska, Maciej Żechowicz, Sylwia Leśniewska, Karolina Osowiecka, Leszek Gromadziński

Round 2
Reviewer 1 Report
Dear authors,
Thank you for your effort in improving your paper. I would like to apologize for a mistake made in my review: When I wrote "ABMP device used by researchers is able to calculate pulse wave velocity" I mean Central Blood pressure values. So again I question if it has been considered by authors to include that information in this paper to increase novelty?.
Because the paper has two main limitations,
First: some of the conclusions reported are widely known and have been shown in other larger cohort studies.Nevertheless, central blood pressure values in extremely obese patients are not so well studied.
Second: the use of ABPM values in hospitalized patients and the absence of clinical blood pressure values (a basic clinical exploration).
Author Response
Dear anonymous reviewer,
Thank you once again for the time and effort, that you have dedicated to further reviewing our corrected manuscript. We are grateful for your positive comments on our paper and understand the concerns and queries. We have additionally worked on the language and style of the paper as required before. We are glad to have incorporated changes, that met your expectations. At the same time we would like to answer the following questions:
Comment: "When I wrote "ABMP device used by researchers is able to calculate pulse wave velocity" I mean Central Blood pressure values. So again I question if it has been considered by authors to include that information in this paper to increase novelty?. Because the paper has two main limitations,
First: some of the conclusions reported are widely known and have been shown in other larger cohort studies. Nevertheless, central blood pressure values in extremely obese patients are not so well studied".
Response: Thank you for this valuable remark. Unfortunately we have not collected data regarding central blood pressure (CBP) values from the beginning, but it is a very good hint for the future examinations of the bariatric patients of course. We agree, that it would increase the novelty of the paper even more. We strongly agree, that the conclusions reported have been shown in other larger cohort studies, but some 'minor' observations were not that obvious and clear. F.e. we have observed, that in the more obese group the diastolic pressure increases more (p<0.001) than systolic pressure as compared to the less obese group. Data on similar dependencies could be only found in small asian groups of patients. "Unlike in our study, according to the paper by Higgins et al. of 1988, based on the Framingham study, greater body weight was associated with increased systolic and diastolic pressure proportionally to baseline". Therefore we have also concentrated on increased diastolic pressure in patients with pathological obesity, that is significantly clinically associated with the pathological alteration in cardiac structure in such patients. And this correlation is strengthened by the non-dipping pattern as well. But we completely agree with your opinion, that adding the CBP in the future analysis could be of great benefit.
Comment: "Second: the use of ABPM values in hospitalized patients and the absence of clinical blood pressure values (a basic clinical exploration)."
Response: Thank you for pointing out this limitation once again, we are sorry not to have given the satisfying response before ("We did not consider clinical measurements on admission as the values ​​would be overestimated by the white coat effect"). Additionally "the benefit of ambulatory 24-hour monitoring over clinical measurements was also evidenced in the context of mortality risk assessment [1]" that was of our interest. According to ESH it is recommended to perform ABPM outside the hospital, but in this study we had only in-hospital measurements technically possible, as the patients were travelling from all over the country. To obey the rule of repeatedly similar conditions for BP measurements on one type of equipment, we have performed them during hospitalisations. At the same time please note that in the study "the impact of obstructive sleep apnoea on the obtained results was excluded (detected in 11.8% of patients in group 1 and 7.4% in group 2, p=0.68) and additionally, a possible impact of sleeping pills on night-time measurements was also excluded (only 2 patients in group 2 took a sleep-inducing drug)".
We hope, that our clarifications and the minor changes in the manuscript meet your expectations. We have thoroughly gone through the limitations of the paper once again and we are aware of all your remarks for the future studies. We look forward to hearing from you in due time regarding our submission and to respond to any further questions and comments you may have.
Yours sincerely,
Beata Moczulska, Maciej Żechowicz, Sylwia Leśniewska, Karolina Osowiecka, Leszek Gromadziński
[1] Dolan E, Stanton A, Thijs L, Hinedi K, Atkins N, McClory S, et al. (2005)Superiority of ambulatory over clinic blood pressure measurement in predicting mortality. Hypertension. 46: 156–161.
Reviewer 2 Report
Well done on addressing my comments and improving the quality of this paper.
Author Response
Dear anonymous reviewer,
Thank you once again for the time and effort, that you have dedicated to further reviewing our corrected manuscript. We are grateful for your newest positive comments on our paper. We are glad to have incorporated changes, that met your expectations. We have additionally worked on the language and style of the paper as required.
Yours sincerely,
Beata Moczulska, Maciej Żechowicz, Sylwia Leśniewska, Karolina Osowiecka, Leszek Gromadziński